# Interactions between Fermi polarons in monolayer WS₂

Jack B. Muir [1,2], Jesper Levinsen [3,4], Stuart K. Earl [1,2], Mitchell A. Conway[1,2], Jared H. Cole [5,6], Matthias Wurdack [7,8], Rishabh Mishra [1,2], David J. Ing[6], Eliezer Estrecho [7,9], Yuerui Lu [10,11], Dmitry K. Efimkin [3,4], Jonathan O. Tollerud[1], Elena A. Ostrovskaya [7,9], Meera M. Parish [3,4] & Jeffrey A. Davis [1,2] ✉

Interactions between quasiparticles are of fundamental importance and ultimately determine the macroscopic properties of quantum matter. A famous example is the phenomenon of superconductivity, which arises from attractive electron-electron interactions that are mediated by phonons or even other more exotic fluctuations in the material. Here we introduce mobile exciton impurities into a two-dimensional electron gas and investigate the interactions between the resulting Fermi polaron quasiparticles. We employ multi-dimensional coherent spectroscopy on monolayer WS₂, which provides an ideal platform for determining the nature of polaron-polaron interactions due to the underlying trion fine structure and the valley specific optical selection rules. At low electron doping densities, we find that the dominant interactions are between polaron states that are dressed by the same Fermi sea. In the absence of bound polaron pairs (bipolarons), we show using a minimal microscopic model that these interactions originate from a phase-space filling effect, where excitons compete for the same electrons. We furthermore reveal the existence of a bipolaron bound state with remarkably large binding energy, involving excitons in different valleys cooperatively bound to the same electron. Our work lays the foundation for probing and understanding strong electron correlation effects in two-dimensional layered structures such as moiré superlattices.

Fermi polarons—mobile impurities immersed in a Fermi gas—provide a versatile model system in which to explore the nature and dynamics of quasiparticles. Here, the impurities become coherently dressed by density fluctuations of the surrounding medium, thus modifying properties such as the impurity mass and lifetime[1,2]. Polaron quasiparticles play an important role in a variety of systems ranging from dilute atomic gases[1] to doped semiconductors[3] to the inner crust of neutron stars[4]. In particular, they have gained much attention in the

[1]Optical Sciences Centre, Swinburne University of Technology, Hawthorn, VIC 3122, Australia. [2]ARC Centre of Excellence in Future Low-Energy Electronics Technologies, Swinburne University of Technology, Hawthorn, VIC 3122, Australia. [3]ARC Centre of Excellence in Future Low-Energy Electronics Technologies, Monash University, Clayton, VIC 3800, Australia. [4]School of Physics and Astronomy, Monash University, Clayton, VIC 3800, Australia. [5]ARC Centre of Excellence in Future Low-Energy Electronics Technologies, RMIT Uinversity, Melbourne, VIC 3001, Australia. [6]Chemical and Quantum Physics, School of Science, RMIT University, Melbourne, VIC 3001, Australia. [7]ARC Centre of Excellence in Future Low-Energy Electronics Technologies, The Australian National University, Canberra, ACT 2601, Australia. [8]Department of Quantum Science and Technology, Research School of Physics, The Australian National University, Canberra, ACT 2601, Australia. [9]Research School of Physics, The Australian National University, Canberra, ACT 2601, Australia. [10]School of Engineering, College of Engineering and Computer Science, The Australian National University, Canberra, ACT 2601, Australia. [11]ARC Centre of Excellence for Quantum Computation and Communication Technology, The Australian National University, Canberra, ACT 2601, Australia. ✉e-mail: JDavis@swin.edu.au

context of ultracold atomic Fermi gases, where experiments have precisely characterized the behavior of Fermi polarons[5–7], including the dynamics of their formation and decay[8]. However, a crucial milestone still missing in cold-atom experiments is a demonstration of polaron–polaron interactions, which are key to understanding many-body phenomena in complex quantum systems, such as itinerant ferromagnetism[9] and unconventional Cooper pairing in superconductors[10].

Recently, atomically thin semiconductors have emerged as a promising platform for studying Fermi polaron quasiparticles[3,11,12]. Specifically, monolayer transition metal dichalcogenides (TMDCs) feature tightly bound excitons (electron-hole pairs), which can be precisely introduced into the system via direct optical transitions. In the presence of excess charges (e.g., electrons), they form two types of exciton-polaron quasiparticles: the attractive polaron, which evolves into a trion (an exciton-electron bound state)[13–16] at low doping; and the repulsive polaron, which becomes the bare exciton at vanishing charge density. By strongly coupling attractive exciton-polarons to cavity photons, it has been experimentally demonstrated that the interactions between polaron-polaritons can be much stronger than those between bare polaritons[17]. However, questions remain about the role of the strong light-matter coupling in these measurements[17], and there is as yet no demonstration of the interactions between matter-only Fermi polarons.

In this paper, we investigate exciton-polarons using multi-dimensional coherent spectroscopy (MDCS), a nonlinear optical technique that has been optimized to probe interactions in quantum matter[18–22]. In monolayer TMDCs, MDCS has previously quantified the exciton and trion coherence times[23–27], as well as the binding energies of biexcitons (bound states of two excitons)[28,29]. Here we reveal the

nature of polaron–polaron interactions in monolayer WS$_2$ by taking advantage of the characteristic spin-orbit splitting of the conduction band at the energetically degenerate K and K′ points[30], as shown in Fig. 1a. We probe the different types of attractive polarons that evolve from trions with distinct spin/valley configurations of excitons and electrons—giving rise to a singlet-triplet energy splitting[14,31]—and we show evidence that interactions only occur between polarons dressed by the same Fermi sea of electrons. Detailed modeling of the polaron physics and the MDCS experiments reveal that these interactions arise from a competition between excitons for the same electrons, as depicted in Fig. 1b. For the case of unbound polarons, this corresponds to a phase-space filling effect which leads to a repulsive interaction between attractive polarons. However, remarkably, our experiment also shows clear signatures of a robust bipolaron state (i.e., a bound pair of attractive polarons) which involves excitons in opposite valleys cooperatively bound to the same electron. Our work thus indicates that two polaron states feature attractive and repulsive branches, similarly to the single-polaron case[3], and sheds new light on the nature of quasiparticle interactions.

## Results
### Polaron–polaron interactions
To determine the nature of the interactions between exciton-polarons, we exploit the underlying trion fine structure in monolayer WS$_2$ at low electron doping. Previous calculations[32–34] and measurements[15,35,36] have obtained trion binding energies within the range 18–35 meV for different monolayer TMDCs. For WS$_2$ and WSe$_2$ monolayers, the conduction band ordering at the K and K′ valleys means excess electrons due to doping occupy the lower energy conduction bands at K and K′ points, while optically accessible (bright) excitons involve the higher

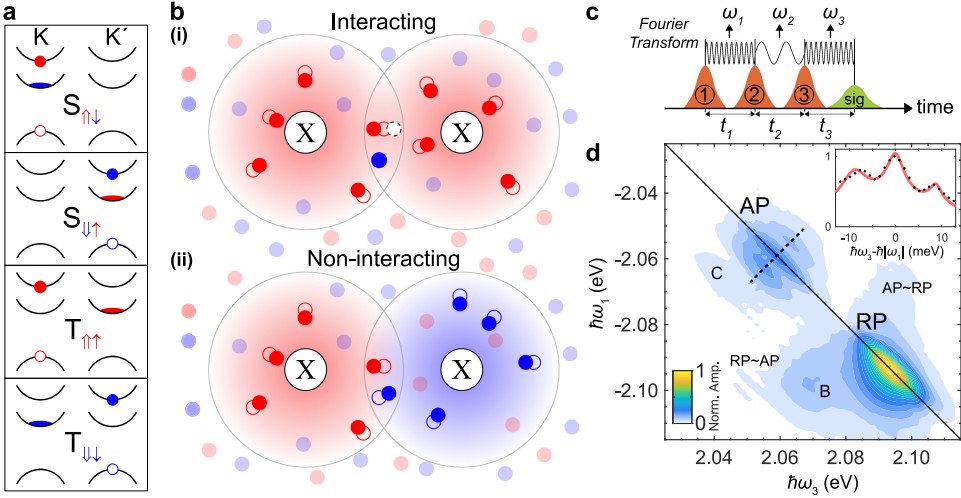

**Fig. 1 | Characterizing polaron states and their interactions in monolayer WS$_2$.** **a** The band ordering in monolayer WS$_2$ gives rise to four distinct attractive polaron states, which we label by the associated singlet and triplet configurations. An exciton consisting of an electron (solid circle) and hole (empty circle) can be dressed by a Fermi sea in the lower conduction band (depicted as flattened ovals). The red (blue) coloring indicates spin-up (-down) electrons. **b** Schematic depiction of attractive polarons and the interactions observed. In (i), two excitons (X) are dressed by the same Fermi sea of ↑ electrons (solid red circles), which are displaced from their equilibrium positions (empty red circles). Between the excitons, an ↑ electron is interacting with the left exciton, making it unavailable to interact with the exciton on the right (empty white dashed circle). This phase-space filling effect leads to an effective repulsive interaction between the two polarons dressed by the same Fermi sea. By contrast, in (ii) the polarons involve distinct Fermi seas, and thus there are no phase-space filling effects and the polarons do not interact via the medium. **c** The pulse ordering in a 1Q rephasing

MDCS experiment, and time periods between them. The fast phase oscillations during time periods $t_1$ and $t_3$ arise from coherent superpositions between states separated by optical photon energies, while phase oscillations during $t_2$ arise from coherences involving states that are much closer in energy. **d** Co-linearly polarized 1Q rephasing 2D spectrum at $t_2 = 0$ is obtained by Fourier transforming the full complex signal measured as a function of $t_1$ and $\omega_3$, and correlates the absorption energy ($\hbar\omega_1$) with the nonlinear signal emission energy ($\hbar\omega_3$). Color scaling represents the normalized absolute signal amplitude (Norm. Amp.). With co-linear polarization, all resonant states are excited and all permitted interactions are present. Several peaks can be identified and attributed to the attractive polaron (AP), repulsive polaron (RP), biexciton (B), bipolaron (C), and interactions between AP and RP (AP - RP and RP - AP). The inset shows a slice perpendicular to the diagonal (indicated by dashed line), clearly showing the splitting of the attractive polaron peaks.

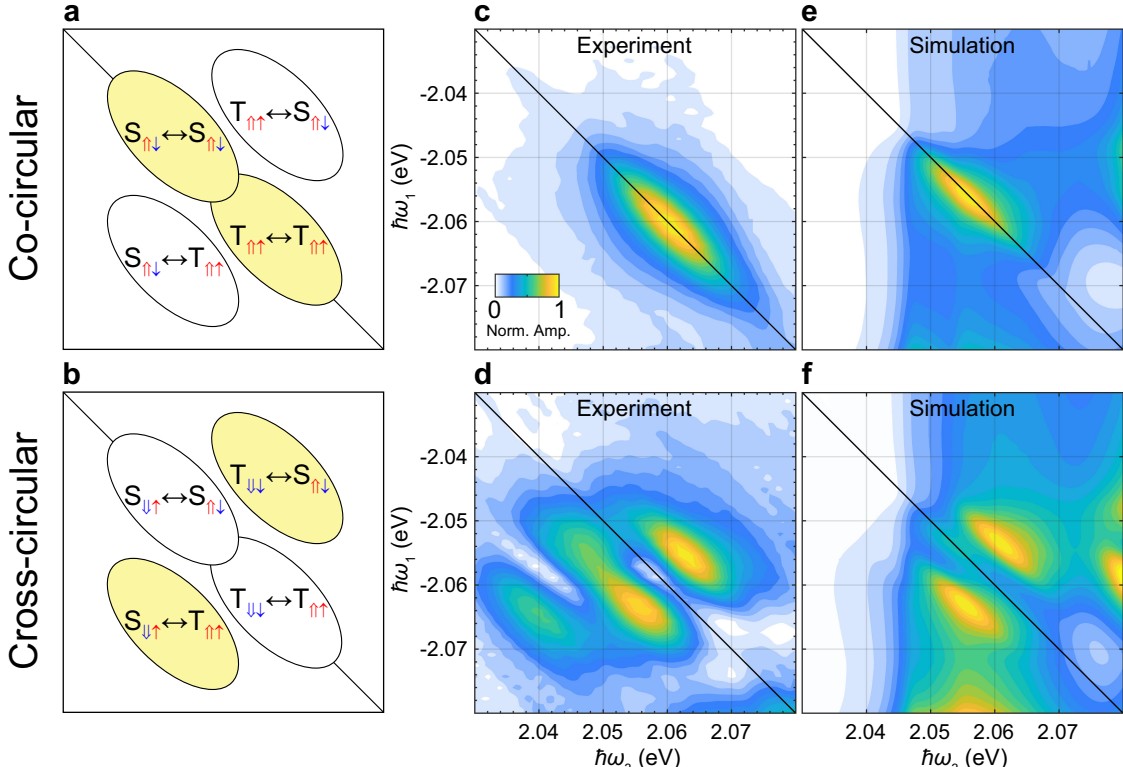

**Fig. 2 | 2D spectra with polarization control reveal interactions between specific polaron states.** Illustrative 2D spectra identify the expected peak locations arising from interactions between specific pairs of attractive polarons in the co-circular (**a**) and cross-circular (**b**) polarization configurations. The detailed pathways that identify the interactions underlying these peaks are described in Supplementary Note 4. The yellow shaded ellipses indicate where peaks are observed in the experiments, and thus which pairs of polarons are interacting. Experimental 1Q rephasing 2D spectra in the vicinity of the attractive polaron peaks for co- (**c**) and cross-circular (**d**) polarization schemes, with $t_2 = 0$. Color scaling represents the normalized absolute signal amplitude (Norm. Amp.). Note:

the cross-circular polarized scheme involves two indistinguishable pulse sequences: $\sigma^+\sigma^-\sigma^+\sigma^-$ and $\sigma^+\sigma^+\sigma^-\sigma^-$. Inhomogeneous broadening causes the on-diagonal peaks to merge, while the cross-peaks can be distinguished due to their narrower homogeneous linewidth. Only diagonal peaks are seen for co-circular polarization, and only cross-peaks for cross-circular polarization. There are additional cross-peaks in **d** at lower $\hbar\omega_3$ values, which are attributed to pathways involving bipolarons. Simulated 2D spectra for the co- (**e**) and cross-circular (**f**) polarization schemes. The simulations reproduce the selection rules for the peaks, confirming that interactions between polarons dressed by the same Fermi sea are dominant and driven by phase-space filling effects.

energy ones. This gives rise to two different trion configurations consisting of an electron in the same or opposite valley as the exciton (Fig. 1a), corresponding to spin singlet (S) and triplet (T) configurations, respectively, which are energetically split by 7 meV in monolayer WS₂[14,37,38] and WSe₂[39,40]. There are then four distinct trion states, labeled $T_{\Uparrow\uparrow}$, $S_{\Uparrow\downarrow}$, $T_{\Downarrow\downarrow}$ and $S_{\Downarrow\uparrow}$, where $\Uparrow$ ($\Downarrow$) represents the pseudospin[41] of an exciton in the K (K′) valley, and $\uparrow$ ($\downarrow$) represents a spin-up (-down) electron, as depicted in Fig. 1a. The trion spin-splitting is inherited by the attractive polarons at larger doping, allowing us to deduce the dominant polaron–polaron interactions in Fig. 1b, as we discuss below. The exfoliated WS₂ monolayer used in this experiment is intrinsically doped due to sulfide vacancies[42] with an electron density estimated to be $1 \times 10^{11}$ cm⁻² (see Supplementary Note 1), which is ~8 times greater than the exciton density, making the polaron framework appropriate.

The MDCS measurements reveal interactions by probing the third-order susceptibility through a phase-sensitive transient four-wave mixing (FWM) experiment (refer to Methods for details). Heterodyne detection ultimately enables measurement of the amplitude and phase of the signal electric field $E(t_1, t_2, \omega_3)$ (see Fig. 1c for time delay definitions), which can be Fourier transformed to give the signal as a function of the different frequencies: e.g., $E(\hbar\omega_1, t_2 = 0, \hbar\omega_3)$. This response can be plotted as a two-dimensional (2D) spectrum and effectively correlates the initial absorption at $\hbar\omega_1$, with the emission of the nonlinear signal at $\hbar\omega_3$. The presence of peaks in such a 2D spectrum is indicative of interactions[19,43] (see Supplementary Note 4 for a detailed discussion).

Figure 1d shows the one-quantum (1Q) rephasing 2D spectrum taken using co-linearly polarized pulses, which allows excitation of all resonant states with each pulse. There are two distinct peaks on the diagonal (where $-\hbar\omega_1 = \hbar\omega_3$), indicating interactions between identical or energetically degenerate quasiparticles. Specifically, the peaks at $-\hbar\omega_1 = \hbar\omega_3 = 2.09$ and 2.06 eV correspond to the repulsive and attractive polarons, respectively, consistent with the peaks we observe in absorption and photoluminescence spectra (see Supplementary Note 1). Both peaks are elongated along the diagonal due to inhomogeneous broadening[21,23], which arises primarily from dielectric disorder[44]. Furthermore, cross-peaks at $(-\hbar\omega_1, \hbar\omega_3) = (2.09, 2.06)$ and $(2.06, 2.09)$ eV are indicative of interactions between the attractive and repulsive polarons. Similar peaks were observed in MDCS measurements on MoSe₂[28]. Finally, a cross-peak due to the biexciton[28,29] is also seen at (2.09, 2.07) eV. The focus of this work, however, is on the peaks in the region of the attractive polaron diagonal peak.

At the energy of the attractive polaron in Fig. 1d, inhomogeneous broadening causes the spin-split singlet and triplet resonances to merge along the diagonal. However, the narrow anti-diagonal linewidth allows us to resolve cross-peaks shifted above and below the diagonal by 7 meV, the singlet-triplet splitting[37,38]. These peaks reveal interactions between different types of attractive polarons. The elongation of the cross-peaks parallel to the diagonal indicates that the interacting polarons experience the same dielectric environment[45]. In this spectral region, there is also another pair of cross-peaks (C) at lower $\hbar\omega_3$ values. As discussed below, we attribute these to a

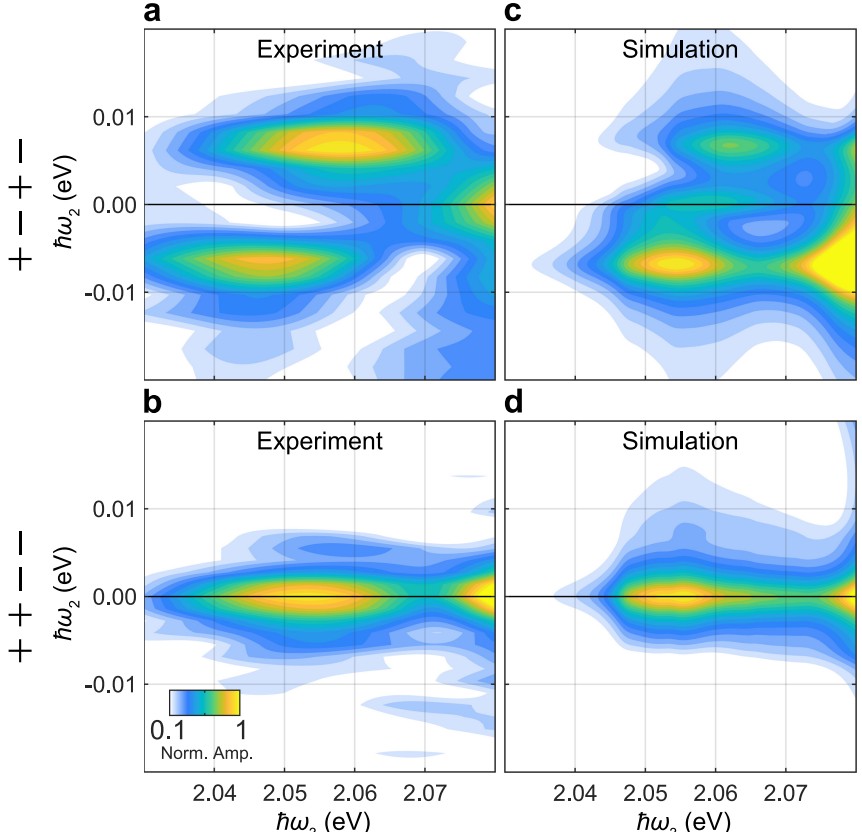

**Fig. 3 | Zero-quantum 2D spectra confirm signal origins. a, b** Experimental 0Q 2D spectra with $t_1 = 50$ fs for the $\sigma^+\sigma^-\sigma^+\sigma^-$ and $\sigma^+\sigma^+\sigma^-\sigma^-$ polarization sequences. Color scaling represents the normalized absolute signal amplitude (Norm. Amp.). (Plots over the full energy range are shown in Supplementary Fig. 5). The peaks at $\hbar\omega_2 = \pm 7$ meV in **a** indicate that in this case the system evolves via the $|T_{\Uparrow\uparrow}\rangle\langle S_{\Downarrow\uparrow}|$ and $|S_{\Uparrow\downarrow}\rangle\langle T_{\Downarrow\downarrow}|$ coherences in $t_2$. In **b** there is only a peak at $\hbar\omega_2 = 0$, as expected for the population pathways driven by the $\sigma^+\sigma^+\sigma^-\sigma^-$ pulse sequence, detailed in Supplementary Fig. 6. In both cases the peaks extend below the singlet and triplet polaron

energies, indicating that the pathways leading to the bipolaron peaks also proceed via these coherences for the $\sigma^+\sigma^-\sigma^+\sigma^-$ case, and via population states for the $\sigma^+\sigma^+\sigma^-\sigma^-$ case. **c, d** The corresponding simulated 2D spectra using the polaron model. These peaks do not extend to low $\hbar\omega_3$ values like they do in the experimental results because the bipolaron states were not included in the model. In both the experiments and simulations, the amplitude of the peaks for $\sigma^+\sigma^+\sigma^-\sigma^-$ and $\sigma^+\sigma^-\sigma^+\sigma^-$ pulse sequences and the associated pathways are roughly equal, providing further validation of the model.

bipolaron, a bound pair of polarons which smoothly evolves into a charged biexciton (an exciton-exciton-electron bound state) at low doping.

To investigate the interactions responsible for the fine structure of the attractive polaron peaks, we use circularly polarized pulses to selectively excite specific pathways. Direct optical transitions at the K and K′ points are permitted only for $\sigma^+$ and $\sigma^-$ circularly polarized light, respectively, enabling valley selective excitation[46,47]. In particular, pulse sequences with identical polarization (co-circular) and orthogonal polarizations (cross-circular) will lead to diagonal and/or cross-peaks only if interactions between specific pairs of states are present, as shown in Fig. 2a, b, and detailed through the full pathway analysis (see Supplementary Fig. 6) and discussion in Supplementary Note 4.

Figure 2c, d shows the 2D spectra measured with co- and cross-circularly polarized laser pulses, respectively. For co-circular polarization, only an elongated diagonal peak is observed. By contrast, for the cross-circular polarized pulses, only cross-peaks are observed, including the two additional peaks associated with the bipolaron. The presence of peaks on the diagonal for the co-circularly polarized excitation is indicative of strong interactions between identical polarons, while the absence of any cross-peaks implies there are no strong interactions between polarons consisting of identical excitons but dressed by electrons in opposite valleys. For the cross-circularly polarized pulses, the presence of cross-peaks and absence of diagonal peaks identifies strong interactions between polarons consisting of excitons in opposite valleys only when the interacting polarons are

dressed by the same Fermi sea. Combined, these results show that regardless of the exciton valley there are strong interactions between polarons only when they are dressed by the same Fermi sea of electrons.

While the location of the peaks clearly reveals which polarons are interacting, understanding the interaction mechanisms requires closer consideration of the pathways generating the signal. In the cross-polarized experiments in Fig. 2d, where $t_2 = 0$, there are two possible pulse orderings with different polarization configurations that cannot be distinguished: $\sigma^+\sigma^-\sigma^+\sigma^-$ and $\sigma^+\sigma^+\sigma^-\sigma^-$. In the former, a coherent superposition between different attractive polaron states (i.e., $|T_{\Uparrow\uparrow}\rangle\langle S_{\Downarrow\uparrow}|$ or $|S_{\Uparrow\downarrow}\rangle\langle T_{\Downarrow\downarrow}|$) is generated by the first two pulses. On the other hand, for the $\sigma^+\sigma^+\sigma^-\sigma^-$ polarization configuration, the first two pulses result in an excited or ground-state population (e.g., $|S_{\Downarrow\downarrow}\rangle\langle S_{\Uparrow\downarrow}|$) (see pathway analysis in Supplementary Note 4 for details). To separate the contributions of these pathways, we measured the signal as a function of $t_2$ and Fourier transformed the data to generate the zero-quantum (0Q) 2D spectra shown in Fig. 3. For the $\sigma^+\sigma^-\sigma^+\sigma^-$ configuration (Fig. 3a), peaks at $\hbar\omega_2 = \pm 7$ meV indicate that the phase of the signal evolves in $t_2$ as a result of a coherent superposition involving excitations separated by 7 meV. That is, the system evolves via the $|T_{\Uparrow\uparrow}\rangle\langle S_{\Downarrow\uparrow}|$ and $|S_{\Uparrow\downarrow}\rangle\langle T_{\Downarrow\downarrow}|$ coherences. By contrast, for the $\sigma^+\sigma^+\sigma^-\sigma^-$ configuration, the 0Q 2D spectrum displays a peak at $\hbar\omega_2 = 0$, as expected for the population pathways. These peaks in the 0Q 2D spectra that extend from $\hbar\omega_3 = 2.03–2.07$ eV directly map on to the peaks in Fig. 2d, since there are no other peaks over this $\hbar\omega_3$ range in

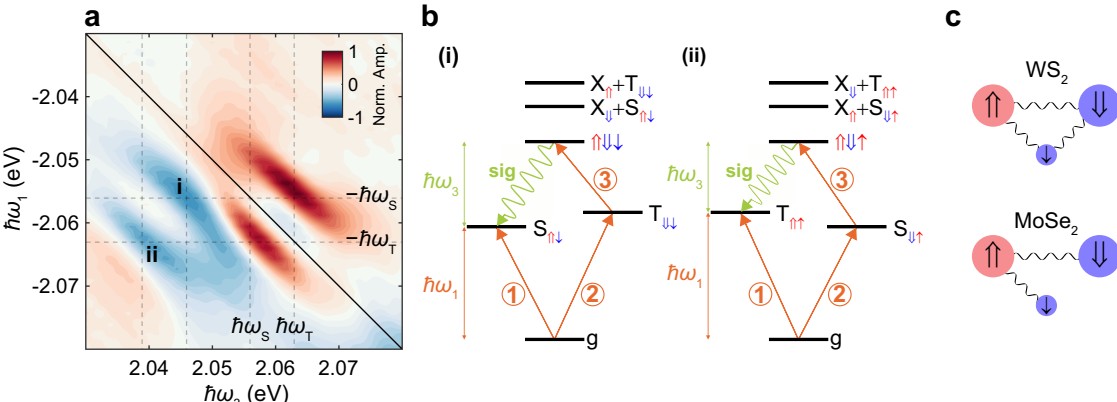

**Fig. 4 | Origin of the bipolaron peaks in the 2D spectrum. a** Real part of the complex 1Q rephasing spectrum from Fig. 2d. Positive peaks arise from stimulated emission pathways (red), while negative peaks appear due to excited state absorption pathways (blue). Color scaling represents the normalized real signal amplitude (Norm. Amp.). S and T absorption/emission energies ($\hbar\omega_S$ and $\hbar\omega_T$, respectively) are marked with dashed lines. The negative peaks arising from the bipolaron are shifted in $\hbar\omega_3$ by 17 meV from $\hbar\omega_S$ and $\hbar\omega_T$. **b** Excited state absorption energy level diagrams showing the pathways which contribute to the i and ii peaks

in **a**. This involves two energetically degenerate bipolarons: ⇑⇓↓ and ⇑⇓↑. The orange arrows indicate laser pulses, while the green sine wave indicates the four-wave mixing signal. **c** Comparison between cooperatively bound charged biexcitons in WS$_2$ monolayers where the participating particles (an electron and two excitons) can all form pairwise bound states, and the case of MoSe$_2$ where trions do not form in the triplet configuration[69]. This explains the larger binding energy in WS$_2$.

either the 0Q or 1Q 2D spectra. The amplitude of the peaks in these 0Q 2D spectra are similar, which suggests that the corresponding pathways are equally probable and contribute roughly equally to the peaks in Fig. 2d, thereby providing a further constraint on modeling of the interactions between polarons dressed by the same Fermi sea.

## Polaron model

To understand the origin of polaron interactions, we have formulated a microscopic model of mobile excitons dressed by Fermi seas in the K and K′ valleys that is appropriate for simulating MDCS spectra (for details, see Methods and Supplementary Note 5). Our model is inspired by polaron theories in ultracold atomic gases[48]. Within our framework, an exciton $X_\sigma$ of pseudospin $\sigma$ is coherently dressed via interactions with electrons, resulting in an effective coupling to the $T_{\sigma\uparrow}$ and $S_{\sigma\downarrow}$ trions. This coupling increases with electron doping, and leads to the formation of repulsive and attractive polaron quasiparticles that are smoothly connected to the underlying exciton and trion states. Interactions between polarons then arise from the fact that the presence of an exciton causes a local depletion of the electron gas available for dressing a second exciton, as depicted in Fig. 1b. Focusing on the unbound polarons that lead to the trion fine structure in Figs. 2 and 3, we model this by assuming that a given electron can dress at most one exciton, thus introducing an effective phase-space filling.

Within our theory, the MDCS spectra are then obtained by calculating the out-of-time-ordered correlators that contribute to the stimulated emission, ground-state bleach, and excited state absorption pathways[49] (Methods). To this end, we consider all states with zero, one, or two excitations, resulting in an effective 28 state Hamiltonian. As opposed to typical treatments of MDCS experiments[18,21], which consider the excitons as two-level systems, we fully incorporate the quantum statistics of excitons, which has previously been shown to be important to explain some features of 2D spectra in GaAs quantum wells[19]. In particular, using this approach, it becomes clear that for the case of non-interacting excitons, the excited state absorption perfectly cancels the ground-state bleach and stimulated emission pathways, leading to zero net signal, as one might expect for a non-interacting system[19,43]. The presence of interactions removes the symmetry of the different pathways and thus the perfect cancellation, leading to a measurable signal.

As seen in Figs. 2 and 3, our model is in excellent agreement with the experimental results. Most notably, it reproduces the interaction selection rules for all polarization sequences in both the 1Q and 0Q 2D spectra. This is particularly remarkable given that the model contains only a minimal set of parameters: the exciton and trion energies, the total exciton and electron densities ($1.3 \times 10^{10}$ cm$^{-2}$ and $1 \times 10^{11}$ cm$^{-2}$, respectively), and the energy scales associated with homogeneous and inhomogeneous broadening. Valley exchange interactions were also considered and included in the model but had minimal impact (see Supplementary Note 5 for details). The strong agreement between the experiment, which clearly identifies the interaction selection rules, and the polaron model, which incorporates minimal free parameters, demonstrates that phase-space filling drives the dominant interactions between unbound attractive polarons in the low electron density regime. This is also consistent with the repulsive interactions observed between polaron-polaritons[17].

## Bipolaron bound state

To further explore the nature of the states contributing to the 2D spectrum in Fig. 2d, we plot its real component in Fig. 4a, which corresponds to the absorptive part of the nonlinear response. Here, we find that the peaks shifted in $\hbar\omega_3$ by -17 meV from the attractive polaron peaks have a negative real part. This indicates that they arise from excited state absorption pathways[19,29] involving a doubly excited state, as depicted in Fig. 4b. The presence of this doubly excited state emitting at these $\hbar\omega_3$ energies is further confirmed by two-quantum MDCS measurements shown in Supplementary Fig. 9. Since these peaks only appear for the cross-polarized pulse sequence, the two excitations making up the doubly excited state are oppositely polarized, as is typically the case for biexcitons[29,50–52]. However, the location of the peaks at $\hbar\omega_1$ values matching the singlet and triplet polaron energies indicates that these attractive polarons are involved in the doubly excited state. Hence we attribute these shifted peaks to a bipolaron state, which is smoothly connected to a cooperatively bound charged biexciton in the limit of low electron density (Fig. 4c).

With $\hbar\omega_3$ = 2.039 and 2.046 eV, these bipolaron peaks occur at similar energy (or shift from the exciton) to peaks in luminescence spectra from monolayer WS$_2$[31,53] and WSe$_2$[50–52] attributed to charged biexcitons. In those measurements, however, there is an ambiguity as to whether the detected emission leaves behind a dark

exciton[50,52] or a dark trion[51,53], which subsequently leads to very different values for the binding energy of any charged biexciton or bipolaron. In the MDCS measurement there is no such ambiguity: all of the excitations involve bright excitons, and it is clear from the pathway analysis (see Supplementary Note 6) that the measured signal arises due to photon emission from the bipolaron leaving behind an attractive polaron (while relaxation to the dark exciton states is possible, the timescale for this relaxation is slow compared to the decoherence time[27]). The difference between the $\hbar\omega_3$ values and exciton energy thus gives binding energies of $53 \pm 1$ meV and $46 \pm 3$ meV, depending on whether the system is left in the $S_{\Uparrow\downarrow}$ or $T_{\Uparrow\uparrow}$ state, respectively (see Fig. 4b). These results are consistent with quantum Monte Carlo calculations for charged biexcitons that treat all electrons and holes as being distinguishable, obtaining a binding energy of 57 meV[54], whereas they cannot be reproduced by theories that model the charged biexciton as an effective two-body (exciton-trion) bound state (which give 14 meV)[34].

The large binding energy can be explained from the $WS_2$ band structure, which allows for a charged biexciton state where all particles are distinguishable. In this case the bound state should not be thought of as a two-body state but rather a three-body bound state where the two excitons and the electron all interact strongly with each other (Fig. 4c). This picture of a cooperatively bound three-body state is further confirmed by the pathways in Fig. 4b (and in Supplementary Note 6) that show that the initial excitation of the doubly excited state proceeds via the $T_{\Downarrow\downarrow}$ ($S_{\Downarrow\downarrow}$) state, but emission is to the $S_{\Uparrow\downarrow}$ ($T_{\Uparrow\uparrow}$) state. Rather than describing this doubly excited state as S (or T) + X, we therefore represent it simply by the pseudospin of the excitons and spin of the electron involved — $\Uparrow\Downarrow\downarrow$ or $\Uparrow\Downarrow\uparrow$, in recognition of the strong interactions between each of them. By contrast, MDCS measurements in $MoSe_2$ have obtained a much smaller charged biexciton binding energy of 5 meV[28]. This is due to the fact that the conduction band ordering is flipped in $MoSe_2$, meaning that the doped electrons lie in the same band as the (bright) excitonic electrons, and thus only interact strongly with one of the excitons (Fig. 4c).

Extending this few-body picture to the polaron framework, the cooperatively bound charged biexciton becomes a bipolaron, where the $S_{\Downarrow\uparrow}$ and $T_{\Uparrow\uparrow}$ polarons are bound via the exchange of excitations in a common Fermi sea of electrons[55]. This is not contained in our minimal polaron model, which is designed to capture the dominant interactions between unbound polarons and thus does not include multi-excitation bound states and their associated peaks in the 2D spectrum (Fig. 2). However, the effective repulsive interactions due to phase space filling are still compatible with the strong attractive interactions in the bipolaron state, since the former occurs for (unbound) polarons separated by large distances compared to the trion size, while the latter occurs at small polaron separation, on the order of the trion size, where the excitons can exchange an electron between each other. Furthermore, this situation resembles the single-polaron case, where underlying attractive exciton-electron interactions give rise to both attractive and repulsive branches. Alternatively, the bipolaron state can be viewed as a biexciton polaron where the biexciton is dressed by the Fermi sea. In this case, however, the internal structure of the biexciton would need to be considered since the coupling to the Fermi sea is larger than the biexciton binding energy, in contrast to the usual exciton-polaron scenario[3,12]. Further measurements as a function of electron density are required to distinguish these different polaron pictures and to understand the role of strong few-body interactions in a many-body system.

## Discussion

Until now, polaron–polaron interactions have proven elusive; however, through the combination of a technique (MDCS) that is optimized to identify interactions and a material (monolayer $WS_2$) that provides fine-structure spitting between polarons, we have been able to unambiguously identify both attractive and repulsive interactions between polarons.

These results demonstrate the ability of MDCS to reveal interactions that are otherwise difficult to disentangle, and open the door to a range of future measurements. In particular, varying the electron density and exciton-electron density imbalance will provide further insights and tests of the models for polaron interactions[2]. Additional details also lie in the shape and linewidths of the two-dimensional peaks, and closer examination of the AP - RP cross-peaks should also reveal details of interactions between attractive and repulsive polarons.

Beyond monolayer TMDCs, the demonstrated ability of these MDCS techniques to disentangle interactions has great potential to provide new insights into the correlated electron physics in moiré superlattices[56]. In these systems, the strength of competing interactions can be tuned[57], and small changes can lead to vastly different macroscopic properties. The results presented here suggest that MDCS is well suited to separating the competing interactions and understanding their interplay, which is essential to unravel the physics of strongly correlated electron systems and to identify means to control their remarkable macroscopic properties.

## Methods

### Sample preparation

The $WS_2$ monolayers investigated in this work were prepared onto gel-films (sourced from gelpak.com) by using the mechanical exfoliation method[58] to thin out the $WS_2$ crystals (sourced from hqgraphene.com). It is essential to remove the bulk material around the monolayer region of interest to prevent scattered light obscuring the signal. This was achieved by using a sacrificial polypropylene carbonate (PPC) film, which is a polymer with a low thermal stability and a glass transition at around 40 °C[59] commonly used for the stacking of van der Waals materials[60]. To accomplish this, PPC dissolved in anisole (concentration of 5%) was initially spin-coated for 1 min at a rotational speed of 3000 rpm on top of a polydimethylsiloxane (PDMS) stamp (sourced from gelpak.com), stabilized with a glass slide, and subsequently baked at 100 °C for 5 min.

The exfoliated $WS_2$ flake was then heated up to 50 °C with an in-house built transfer setup to pick up the bulk material around the monolayer with the PDMS/PPC stamp. This was achieved by aligning a clean area of the stamp with the bulk material and slowly lowering it. Importantly, the stamp was just brought in contact with the bulk, and not with the monolayer. The direction from which the stamp was attaching to the Gel-film was adjusted with a tilt-control of the transfer stage. After bringing the stamp in contact with the bulk material, the stamp was quickly lifted up and the separation of the bulk material from the monolayer achieved. By repeating this process for all the bulk material surrounding the monolayer, the monolayer was completely isolated.

Finally, to transfer the isolated monolayer onto the $SiO_2$ chip (sourced from novawafers.com), we picked it up with another PPC/PDMS stamp at 50 °C. The transfer was then completed by melting the PPC film with the monolayer onto the target substrate at 120 °C. The PPC residues after the transfer were washed off by subsequent bathing the sample in acetone, isopropanol, and ethanol.

### Multi-dimensional coherent spectroscopy

The MDCS measurements were performed on monolayer $WS_2$ (on Si/ $SiO_2$ substrate) in a cryostat (Montana Instruments—Cryostation) at 6 K and low pressure. Laser pulses with duration 22–24 fs were used, with their spectrum centered at ~2 eV which covered both the attractive and repulsive polaron transitions (see Supplementary Fig. 1). The fluence at the sample was kept below 1 μJ cm$^{-2}$ per pulse, ensuring higher order effects did not contribute to the signal (see Supplementary Fig. 4). The pulses were selected at a repetition rate of 12.5 kHz.

These pulses were generated in a Noncolinear Optical Parametric Amplifier (Light Conversion Orpheus-N) pumped by a Yb:KGW laser amplifier (Light Conversion Pharos).

The measurements were done in a box-CARS geometry[21] (see Supplementary Fig. 3) with the four beams generated by focusing the single beam onto a 2D grating and selecting the four first-order diffracted beams. All lenses from this point on were in 4F geometry to ensure precise imaging of the overlapping spot from the 2D grating to the sample. All four beams were incident on common optics to ensure high phase stability[21].

The delays between pulses were introduced using a pulse shaper based on a spatial light modulator (Santec LCOS-SLM-100), which allows control of the spectral amplitude and phase of each beam independently[21,61]. The pulse shaper also enabled pulse compression at the sample position using the multi-photon intra-pulse interference phase scan technique[62], to correct for temporal chirp introduced by the optics and beam propagation.

To control the circular polarization of the initially co-linearly polarized pulses, a half-wave plate (HWP) was placed in the path of pulses $\mathbf{k}_1$ and $\mathbf{k}_3$ to control their linear polarization direction, while a second HWP was placed in the path of pulses $\mathbf{k}_2$ and the local oscillator beam ($\mathbf{k}_{LO}$). A single quarter-wave plate in all beams was then used to convert the linearly polarized pulses to circular polarized. In this case the direction of the linear polarization after the HWPs was selected so that the pulses were were either $\sigma^+$ or $\sigma^-$ polarized, as required for the different pulse polarization sequences.

The four beams in a box geometry were focused through a 75 mm plano-convex lens onto the sample, where they were spatially overlapped, with spot size of 54 μm FWHM (see Supplementary Fig. 1). The FWM signal emitted in the direction $\mathbf{k}_{sig} = -\mathbf{k}_1 + \mathbf{k}_2 + \mathbf{k}_3$ overlaps with the fourth corner of the box (see Supplementary Fig. 3) and $\mathbf{k}_{LO}$, which was attenuated to be 3 orders of magnitude weaker than the other beams, and thus similar to the signal amplitude. The signal and LO beam were sent into a spectrometer with CCD detector that records the resultant spectral interferogram, from which both the amplitude and phase of the signal can be retrieved. The signal was measured as a function of $t_1$ for the 1Q 2D spectra and $t_2$ for the 0Q 2D spectra. Measuring the spectral amplitude and phase of the signal ensures there is a unique solution to the Fourier transform of the data with respect $t_1$, $t_2$, and/or $\omega_3$, thereby allowing generation of the 2D spectra.

## Polaron model and simulations

To model the attractive and repulsive polarons in the low-doping regime, we consider the simplest many-body Hamiltonian that involves excitons, trions, and the coupling between these mediated by Fermi seas in the K and K′ valleys:

$$
\hat{H}_0 = \sum_{\sigma = \Uparrow, \Downarrow} \left( E_X^0 \hat{X}_\sigma^\dagger \hat{X}_\sigma + E_T^0 \hat{T}_\sigma^\dagger \hat{T}_\sigma + E_S^0 \hat{S}_\sigma^\dagger \hat{S}_\sigma \right)
$$
$$
+ \sum_{\sigma = \Uparrow, \Downarrow} \left( \alpha_t \sqrt{N_e} \hat{T}_\sigma^\dagger \hat{X}_\sigma + \alpha_s \sqrt{N_e} \hat{S}_\sigma^\dagger \hat{X}_\sigma + \text{H.c.} \right). \tag{1}
$$

The terms in the first set of parentheses correspond to the bare excitons at energy $E_X^0$ and to the trions with energies $E_S^0$ (singlet) and $E_T^0$ (triplet), where all these energies are in the absence of any exciton-trion coupling. We denote the exciton creation operator $\hat{X}_\sigma^\dagger$, and we distinguish the singlet and triplet operators by the pseudospin of their excitonic component such that, e.g., $\hat{S}_\Uparrow^\dagger$ corresponds to the creation operator of the singlet trion $S_{\Uparrow\downarrow}$. Here it should be understood that the creation of a trion leaves behind a hole in one of the Fermi seas, but since the hole's momentum is effectively zero at low doping, we neglect its dynamics.

The terms in the second set of parentheses in Eq. (1) correspond to the coupling between trions and excitons. This coupling is mediated by the electrons in the K and K′ Fermi seas, where an exciton can bind

to an electron to form a singlet or a triplet trion. At low doping, the strength of the coupling can be approximated as[63]:

$$
\alpha_j \sqrt{N_e} \simeq \sqrt{\frac{2\pi\varepsilon_j N_e}{m_r A}} = \sqrt{\frac{3}{2}} \sqrt{\varepsilon_j E_F}, \tag{2}
$$

where $m_r = m_e m_X/(m_e + m_X) \simeq \frac{2}{3} m_e$ is the reduced electron-exciton mass in terms of the effective electron and exciton masses $m_e$ and $m_X$, $A$ is the area per exciton (i.e., the inverse exciton density), $N_e$ is the number of electrons per valley within this area, $E_F = 2\pi N_e/m_e A$ is the Fermi energy, and $\varepsilon_j$ is the $j$-trion binding energy, i.e., $E_S^0 = E_X^0 - \varepsilon_s$ and $E_T^0 = E_X^0 - \varepsilon_t$. This form of the coupling yields the expected oscillator strengths at low doping for the attractive and repulsive polarons arising from the coupling to a trion state found both in polaron and trion models of the doped semiconductor[63,64]. While it overestimates the energy splitting between the two polaron branches with doping (by a factor of 2), this can be corrected for by including an additional doping dependence in the exciton energy.

We now consider the interactions between polarons. First, we emphasize that the Hamiltonian in Eq. (1) does not contain any direct exciton-exciton interactions, since these are expected to be small for low-momentum scattering[65,66]. Instead, the interactions arise due to doping. To see this, we treat the electrons (and therefore the trions) as effectively localized on the timescale of the experiment. This is reasonable since the Fermi time $\hbar/E_F \simeq 2$ ps exceeds the maximum time delay between the excitation pulses, and hence the dynamics of the Fermi sea occurs only at time scales beyond those probed in the experiment. At the same time, the exciton momentum is much smaller than any other relevant scale in the system. Therefore, the (effectively) zero-momentum delocalized exciton couples to a coherent superposition of (effectively) localized trions on different sites within the area $A$, such that:

$$
\hat{S}_\sigma = \frac{1}{\sqrt{N_e}} \sum_{j=1}^{N_e} \hat{S}_{\sigma,j}, \qquad \hat{T}_\sigma = \frac{1}{\sqrt{N_e}} \sum_{j=1}^{N_e} \hat{T}_{\sigma,j}, \tag{3}
$$

where $j$ labels the position of an electron (which is transformed into a trion via the exciton). To obtain the interactions, we then use the fact that there can only be one trion per electron, giving:

$$
(\hat{S}_{\sigma,j})^2 = (\hat{T}_{\sigma,j})^2 = \hat{S}_{\Uparrow,j} \hat{T}_{\Downarrow,j} = \hat{T}_{\Uparrow,j} \hat{S}_{\Downarrow,j} = 0. \tag{4}
$$

This demonstrates that interactions arise effectively due to phase-space filling. On the other hand, trions that involve electrons of different spins (and hence different sites) are not affected by phase-space filling, and thus terms such as $\hat{S}_{\sigma,j} \hat{T}_{\sigma,j}$ are non-zero. For further details on the induced interactions, see Supplementary Note 5.

To model the MDCS results, we note that the signal contributing to the 0Q and 1Q rephasing experimental protocols originates from three processes[49]: Ground-state bleach, stimulated emission, and excited state absorption. In order, their contributions to the material polarization take the form:

$$
P_{\sigma\sigma'}^{(3)} = -\left(\frac{i}{\hbar}\right)^3 \mu_X^4 E_0^3 \left( \text{Tr}\left[ \hat{\rho}_0 \hat{X}_\sigma e^{i\hat{H}_0 t_1} \hat{X}_\sigma^\dagger e^{i\hat{H}_0(t_2+t_3)} \hat{X}_\sigma e^{-i\hat{H}_0 t_3} \hat{X}_{\sigma'}^\dagger \right] \right.
$$
$$
+ \text{Tr}\left[ \hat{\rho}_0 \hat{X}_{\sigma'} e^{i\hat{H}_0(t_1+t_2)} \hat{X}_\sigma^\dagger e^{i\hat{H}_0 t_3} \hat{X}_\sigma e^{-i\hat{H}_0(t_2+t_3)} \hat{X}_\sigma^\dagger \right]
$$
$$
\left. - \text{Tr}\left[ \hat{\rho}_0 \hat{X}_{\sigma'} e^{i\hat{H}_0(t_1+t_2+t_3)} \hat{X}_\sigma e^{-i\hat{H}_0 t_3} \hat{X}_{\sigma'}^\dagger e^{-i\hat{H}_0 t_2} \hat{X}_\sigma^\dagger \right] \right). \tag{5}
$$

Here, the initial state density matrix $\hat{\rho}_0$ describes the two Fermi seas in the K and K′ valleys in the absence of excitons, $\mu_X$ is the exciton dipole moment, and $E_0$ is the electric field strength. The co- and cross-polarized cases correspond to $\sigma = \sigma'$ and $\sigma \neq \sigma'$, respectively.

Importantly, in the absence of polaron interactions, the form of Eq. (5) implies that the excited state absorption precisely cancels the ground-state bleach and stimulated emission pathways.

The modeling furthermore incorporates both homogeneous and correlated inhomogeneous broadening (see Supplementary Note 5) which were quantified by fits to the experimental data (see Supplementary Note 7). We have checked that all theory results presented are well reproduced by a Lindblad master equation approach[67,68] (see Supplementary Fig. 7).

## Data availability

The experimental data used to generate the 2D spectra in this study are available in figshare under the accession code: https://doi.org/10.6084/m9.figshare.21215231.

## Code availability

The source code for the simulations is available from the corresponding author upon reasonable request.

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

## Acknowledgements

This work was supported by the Australian Research Council Center of Excellence in Future Low-Energy Electronics Technologies (CE170100039). M.M.P. and J.L. were supported through the Australian Research Council Future Fellowships FT200100619 and FT160100244, respectively. J.H.C. and D.J.I. also acknowledge the support of the Australian National Computational Infrastructure (NCI) and the ARC Centre of Excellence in Exciton Science (CE170100026). Y.L. acknowledges the support of ARC Centre of Excellence in Quantum Computation and Communication Technology (CE170100012). E.E. was supported through the Australian Research Council Discovery Early Career Research Award DE220100712.

## Author contributions

J.A.D. conceived and supervised the project. M.W. fabricated the sample and with E.E. performed the physisorption gating measurements under the supervision of E.A.O. and Y.L. J.B.M., M.A.C., and S.K.E. conducted the MDCS measurements with assistance from J.O.T. and R.M. The data were analyzed and interpreted by J.B.M., J.A.D., and S.K.E., with input from J.O.T., M.A.C., J.H.C., M.W., E.A.O., J.L., and M.M.P. M.M.P., J.L., and D.K.E. developed the polaron model, including phase space filling effects. J.L. and M.M.P. developed the MDCS simulation with input from J.A.D., J.H.C., and J.B.M. D.J.I. and J.H.C. developed the simulations using the Master Equations approach. J.B.M., J.L., M.M.P., and J.A.D. wrote the paper with input from all authors.

## Competing interests

The authors declare no competing interests.
