## [Peer Review File · Nature Communications]

Reviewers' Comments:

Reviewer #1:

Remarks to the Author:

Bose-polarons and Fermi-polarons are subject to intense experimental and theoretical research and the understanding of polaron-polaron interaction and the bipolaron structure is certainly a very important research direction. The authors use TMDCs which are particularly suited to the study of polaron-polaron interactions because one can create excitons in a controlled way and one can select the desired type of polarons by doping.

The paper is well written, clear, and thorough. The theoretical part uses a simple model and the good agreement with the experiments is somewhat surprising. But given the difficulty of the theoretical description of such a complex system, there is no suitable alternative approaches.

Minor remarks:

1. 'emission leading to the signal leaves the system in an attractive polaron state'

Do you mean attractive polaron plus exciton state?

2. Fig 4c is not very convincing, I wonder if it is really needed. The Coulomb interaction is long-ranged even in these materials and I do not think the band structure argument is strong enough.

In overall, I have found this paper interesting and important contribution and recommend it for publication.

Reviewer #2:

Remarks to the Author:

In this manuscript, the authors report measurements of exciton-polaron interactions in monolayer WS₂ by multi-dimensional coherent spectroscopy. The results could be potentially useful to understand the complex interactions in TMDs. Before recommending this manuscript, I hope the authors could address the comments below.

1. What could be different if using the trion model to replace the polaron in the analysis? The Trion model is already good enough at low carrier density.
2. On page6, "the narrow anti-diagonal linewidth allows us to resolve cross-peaks shifted above and below the diagonal by 7meV, the singlet-triplet splitting", it's better to show the crosscut line for the cross-peaks.
3. The two factors are usually considered for the polaron at low doping: screening and phase-filling effect [PRB 95, 035417 (2017)], and the phase-space filling effect is the dominant factor for the interactions between polarons, this has been shown with different methods [Nature Communications 12, 6131 (2021)].
4. Can the authors have a brief discussion about the possible influence of dark excitons and trions on the interaction of polarons?
5. The transfer method has been used during the preparation of the sample, why don't use BN to encapsulate the monolayer, that could make the linewidth much smaller, and the signal should be improved.

Reviewer #3:

Remarks to the Author:

The manuscript by Muir et al. reports multi-dimensional coherent spectroscopy (MDCS) study of monolayer WS₂. Distinct signals between co- and cross polarized configurations are observed,

indicating that interactions only happen between specific combinations of attractive polarons. The MDCS spectra also show additional features at lower energy, which are assigned to bipolarons. Overall, I find the results interesting. On the other hand, I am not fully convinced by the microscopic pictures in the manuscript.

1. The main question I have is on the sign of interaction. The authors argue that there is a long-range repulsive interaction between e.g. S_{du} and T_{uu} (following the terminology in the manuscript, the first u/d stands for electron spin within the exciton, and the second u/d stands for spin of the fermi sea), and a short-range attractive interaction between them. While the latter is strongly supported by the observation of bipolaron, the former (long-range repulsive interaction) seems to be only supported by the theoretical picture in Fig. 2b without experimental evidence. How are short- and long-range interaction distinguished in the present experiments? It seems that some excitation density dependence would be helpful.

2. Furthermore, I am not sure whether the long-range interaction is necessary or relevant in the first place. The observation of bipolaron emission at ($w_1 = w_s, w_3 = w_t - 17\text{meV}$) and ($w_1 = w_t, w_3 = w_s - 17\text{meV}$) suggest that S_{du} and T_{uu} will attract each other to share the same fermi sea and form a biplaron to lower the total energy by 17meV (Fig. 4b). This already breaks the symmetry of transitions and seems sufficient to generate all features in Fig. 2 and 3 (e.g. Nano Lett. 2016, 16, 8, 5109–5113). Can all experimental observations be accounted for solely by this attractive interaction? Actually, from a naïve picture, signal electric field in Fig. 3 is only contributed by S_{du} and T_{uu} that are spatially close, where short range interaction is likely more relevant (and certainly stronger). As an extreme example, if we separate the sample into two pieces and have E1, E3 only hitting one piece but E2 only hitting the other piece, the signal would be very small since at the end the transition matrix element between ground state in one piece and excited state in the other piece is small.

3. Fig. 2 is very nice and worthy of more discussion. The manuscript states the conclusion with one sentence that “this indicates that interactions between polarons dressed by the same Fermi sea dominate”, which to my opinion is too simplified. For example, Fig. 2 offers strong evidence that interactions only happen between S_{du} and T_{uu} or T_{dd} and S_{ud} , but not between S_{ud} and T_{uu} or T_{uu} and S_{ud} . This conclusion is convincing. However, it can be explained by two scenarios: the fact that S_{ud} and T_{uu} do not interact could be either due to that the “d” fermi sea and the “u” fermi sea have different spins and do not interact, which is the picture used in the manuscript (Fig. 2b); or due to that two “u” exciton (i.e. two excitons from the same valley) strongly repel each other and therefore always stay far apart and do not interact. The absence of $S_{ud} \leftrightarrow S_{du}$ or $T_{ud} \leftrightarrow T_{du}$ in Fig. 2b actually suggests that indeed the interaction is still weak even if two excitons are from the different valleys, as long as the Fermi seas are different (as opposed to the exciton case in Nature Physics 12, 677–682 (2016)). On the other hand, this is still consistent with the fact that only two polarons with the same Fermi sea can form a bipolarons (Fig. 4b), and therefore can be accounted for by solely considering the short-range attractive interaction.

A minor terminology issue: the authors use the first u/d to represent “spin of the exciton”, which is not accurate. All excitons discussed in the manuscript are bright and have total spin 0; the first u/d is representing spin of the electron within the exciton, not the exciton as a whole.

We are grateful for the considerations of the reviewers, which have helped us to improve our manuscript. Below, we address the reviewers' comments point by point (in blue text), and detail all changes to the manuscript (in green).

RESPONSE TO REVIEWER #1

We thank the reviewer for their positive comments and recommendation for publication. Regarding their minor remarks:

1. "emission leading to the signal leaves the system in an attractive polaron state"

Do you mean attractive polaron plus exciton state?

While it is true that the bipolaron state would be breaking into an attractive polaron plus exciton, in this case the exciton has recombined and emitted a photon, which leaves just the attractive polaron. To clarify this in the manuscript we have changed the text to:

"the measured signal arises due to photon emission from the bipolaron leaving behind an attractive polaron."

2. Fig 4c is not very convincing, I wonder if it is really needed.

The Coulomb interaction is long-ranged even in these materials and I do not think the band structure argument is strong enough.

Figure 4c illustrates the qualitative difference between the three-body systems consisting of two excitons and an electron in WS₂ versus MoSe₂: In the case of WS₂, all two-body subsystems can form bound states (either a biexciton or trions), whereas in MoSe₂ only two of these subsystems can form bound states. This is a consequence of the difference in the band structure in the two materials, and these different bound states have been verified in experiment. Hence, the three-body bound state (exciton-exciton-electron) is tighter bound in WS₂ than in MoSe₂.

To address this point, we have added the new reference 55 to the figure caption for Fig. 4c, which explicitly discusses how and why triplet and singlet trions exist in WS₂, whereas triplet trions do not exist in MoSe₂.

In all cases, while the interactions of course originate from the Coulomb interaction, the actual interaction between an electron and an exciton or the interaction between two excitons are short-ranged (involving induced dipoles).

RESPONSE TO REVIEWER #2

We are pleased that the reviewer finds the results to be "potentially useful" and we thank them for their detailed comments, which we address in the following.

1. What could be different if using the trion model to replace the polaron in the analysis? The Trion model is already good enough at low carrier density.

Indeed, it has been shown that approaches based on trions can yield comparable results to polaron models at low carrier density [e.g., Glazov, J. Chem. Phys. 2020]. However, this is only the case for

certain observables (e.g., a trion model can capture the weight of the attractive polaron but not its energy shift). Furthermore, it has not been used to describe the *polaron-polaron interactions* that we investigate in this work, and it is not clear that this can be done. By contrast, a competition for electrons naturally emerges in our polaron model where excitons interact with multiple electrons rather than just being bound to one electron as in a trion. Therefore, the polaron description provides the simplest and most physical way to model the experiment.

We have now added a reference to the trion model [65] in the methods section, below Eq. (2), The sentence now reads:

“This form of the coupling yields the expected oscillator strengths at low doping for the attractive and repulsive polarons arising from the coupling to a trion state found both in polaron and trion models of the doped semiconductor [64, 65]”

2. On page6, “the narrow anti-diagonal linewidth allows us to resolve cross-peaks shifted above and below the diagonal by 7meV, the singlet-triplet splitting”, it’s better to show the crosscut line for the cross-peaks.

We agree and thank the referee for the suggestions. We have now added a cross-diagonal slice to the inset of Fig 1d. The cross-diagonal slices for the data in Fig 2c and d are included in Fig S13 of the Supplementary Information

3. The two factors are usually considered for the polaron at low doping: screening and phase-filling effect [PRB 95, 035417 (2017)], and the phase-space filling effect is the dominant factor for the interactions between polarons, this has been shown with different methods [Nature Communications 12, 6131 (2021)].

These papers and the two factors mentioned by the referee relate to the interactions between electrons and excitons that lead to the formation of polarons. The Nat Comms paper discusses phase space filling as the reason why the attractive polaron amplitude decreases at high doping. This is a different type of effect and arises because the Fermi sea of electrons prevents absorption of light and creation of an exciton. The phase space filling effect we discuss is different and relates to the number of electrons available to interact with the exciton. This leads to interactions between polarons, which is not discussed in those papers.

This point also highlights a potential ambiguity in our title, which may lead to some confusion. We have therefore changed the title to:

“Interactions between Fermi polarons in monolayer WS₂”

which removes any ambiguity and makes clear that we are looking at interactions between polarons.

4. Can the authors have a brief discussion about the possible influence of dark excitons and trions on the interaction of polarons?

In the present work, all of the excitations probed involve bright excitons. Some of the optically excited excitons will relax down to the lower energy dark exciton states, however the timescale for this relaxation is slow (~780 fs [[27] - (Jakubczyk, 2018)]) compared to the timescale of these measurements, which is limited by the decoherence times of the excitons/trions/polarons (<400fs).

Hence, we do not expect dark states to contribute to the measured response in any way [29]. We have modified the following text to the manuscript to make this point:

“In those measurements, however, there is an ambiguity as to whether the detected emission leaves behind a dark exciton [50, 52] or a dark trion [51, 53], which subsequently leads to very different values for the binding energy of any charged biexciton or bipolaron. In the MDCS measurement there is no such ambiguity: all of the excitations involve bright excitons, and it is clear from the pathway analysis (see Supplementary Information) that the measured signal arises due to photon emission from the bipolaron leaving behind an attractive polaron (while relaxation to the dark exciton states is possible, the timescale for this relaxation is slow compared to the decoherence time [27]).”

5. The transfer method has been used during the preparation of the sample, why don't use BN to encapsulate the monolayer, that could make the linewidth much smaller, and the signal should be improved.

hBN encapsulation has indeed been shown to make the inhomogeneous linewidths smaller, which may help in some aspects of interpreting the signal. However, the inhomogeneous broadening that arises from not having encapsulated samples is less important in these measurements where the rephasing pulse ordering effectively removes this broadening in the anti-diagonal direction of the 2D spectra. This allows us to resolve homogeneous linewidths and separate the different trion peaks. Removing the inhomogeneous broadening is not expected to change this.

In these experiments, relatively large area (>30um) samples are required to accommodate the overlapping spot size in the non-collinear geometry used (Fig. S3). The result is that uniform hBN encapsulation over this area is more difficult and the additional layers can lead to increased scatter, making it harder to observe the signal above the noise. We therefore decided to not complicate the fabrication, and perform measurements on a pristine bare WS₂ monolayer. Furthermore, the intrinsic doping that allows us to observe the trion/polaron peaks is enhanced by the SiO₂/Si substrate and is typically quenched to a large extent in encapsulated samples.

RESPONSE TO REVIEWER #3

We are pleased that the reviewer finds the results to be “interesting” and we thank them for their detailed comments, which we address in the following.

1. The main question I have is on the sign of interaction. The authors argue that there is a long-range repulsive interaction between e.g. S_{du} and T_{uu} (following the terminology in the manuscript, the first u/d stands for electron spin within the exciton, and the second u/d stands for spin of the fermi sea), and a short-range attractive interaction between them. While the latter is strongly supported by the observation of bipolaron, the former (long-range repulsive interaction) seems to be only supported by the theoretical picture in Fig. 2b without experimental evidence. How are short- and long-range interaction distinguished in the present experiments? It seems that some excitation density dependence would be helpful.

The repulsive interactions originating from phase space filling effects described in the model are necessary to consistently explain the peaks observed in both the co-circular polarized excitation and cross-circular polarized excitation schemes. In the absence of these interactions there would be no peaks on the diagonal in the 2D spectrum with co-circular polarization, or cross-peaks in the case of cross-circular polarization. The simplest self-consistent explanation that can explain the presence of all peaks, except the bipolaron peaks, and the absence of the other peaks, is via the repulsive interactions arising from phase space filling effects, as described in the model.

To help clarify the point, we note that many of the signals observed in previous MDCS measurements on excitonic systems do, in fact, arise primarily as a result of repulsive interactions. To explain this, one needs to consider that these are bosonic systems, and it is possible to create multiple excitons in the same quantum mechanical state. If we consider just a single type of exciton in this framework, then it can then be shown that in the absence of interactions there will be no signal, even on the diagonal, due to the cancellation of ground state bleach (GSB) + stimulated emission (SE) pathways with the excited state absorption (ESA) pathway [[43]-(Mukamel, 2000)] (an important point to note here is that the bosonic nature of excitons requires that the transition dipole from the 1-exciton state to the 2-identical-exciton state is a factor of $\sqrt{2}$ larger than the transition from 0-exciton to 1-exciton state). This statement that there is no signal in the absence of interactions may seem counterintuitive to many people accustomed to thinking of these experiments in the context of two-level systems (or few-level systems), where simply the presence of a bright state generates a peak on the diagonal. In this type of two-level system, there is no possibility of having two excitons in the same state, no ESA pathway, hence no cancellation, and thus a signal appears on the diagonal. The inability to excite a second exciton is equivalent to having an infinite repulsive interaction between identical excitons. Hence the presence of a signal for a two-level system is consistent with this framework of requiring interactions for peaks to appear.

We have added a summary of these points to become the third paragraph in Supplementary Information Section IV: QUANTUM MECHANICAL PATHWAY ANALYSIS:

“This statement that there is no signal in the absence of interactions, even for diagonal peaks, may seem counter-intuitive to those accustomed to thinking of these experiments in the context of two-level systems (or few-level systems), where simply the presence of a bright state generates a peak on the diagonal. In the two-level system, there is no possibility of having two indistinguishable excitons in the same state, no ESA pathway to cancel the GSB and/or SE, and thus a signal appears on the diagonal. The inability to excite a second exciton is equivalent to having an infinite repulsive interaction between identical excitons. Hence, the presence of a signal for a two-level system is consistent with this framework of requiring interactions for peaks to appear.”

The presence of repulsive interactions, as described through the model, is thus essential to be able to explain the presence and absence of the peaks in the different polarization schemes.

It is possible, as the reviewer suggests below, that the presence of the bipolarons could lead to the polaron cross-peaks observed in the cross-polarized measurements; however, this requires a coupling between the unbound two-polaron states and the bipolarons. While initially it may seem like this coupling should be obvious, the microscopic nature of such coupling is less clear. Unlike the case of biexcitons, previously observed in 2D spectra of semiconductors [[18]-(Karaiskaj 2010), [21]-(Tollerud 2017)], the bound state does not arise simply from the binding of the two fundamental particles. In the few-body picture this would lead to the formation of two-trions bound together, i.e. a 6-body state consisting of 4 electrons and 2 holes. Previous calculations have shown that such a

complex is not stable [[54] - Mostaani, PRB 2007]. Instead, as discussed in the manuscript, the bipolaron evolves into a 5-body state consisting of 3 electrons and 2 holes. The coupling between this state and the unbound 2-polaron state is thus more difficult to describe from a microscopic perspective. In contrast, the phase-space filling that leads to the repulsive interactions and accurately reproduces the experimental peaks is described naturally in the microscopic model, with limited assumptions.

Additionally, the bipolaron peaks are almost a factor of 2 weaker than the polaron cross-peaks. If the sole origin of the asymmetry was the presence of the bipolaron pathway then the amplitudes should be identical.

Regarding the excitation density dependence, it is true that the balance of attractive and repulsive interactions may change as the density changes, although both would be expected to lead to increased signal, at least initially. To quantitatively compare, we would need to have a detailed understanding of the microscopic mechanisms for the bipolaron formation, which is currently not the case and is beyond the scope of this work. Furthermore, the role of the electron density (Fermi level) is also important, making changes to the exciton density only even more complicated.

We have however done intensity dependent measurements over more than an order of magnitude for co-linear excitation (Fig S4). This reveals that all peaks scale as expected for the $\chi^{(3)}$ response. No changes in the amplitude of the bipolaron peak relative to polaron diagonal or cross-peaks are observed. We note, however, that at the highest power the average exciton separation is $>1\mu\text{m}$, much larger than the size of any of the quasiparticles discussed, and far from the regime where short-range interactions may be expected to be enhanced.

2. Furthermore, I am not sure whether the long-range interaction is necessary or relevant in the first place. The observation of bipolaron emission at ($w_1 = w_s, w_3 = w_t - 17\text{meV}$) and ($w_1 = w_t, w_3 = w_s - 17\text{meV}$) suggest that S_{du} and T_{uu} will attract each other to share the same fermi sea and form a bipolaron to lower the total energy by 17meV (Fig. 4b). This already breaks the symmetry of transitions and seems sufficient to generate all features in Fig. 2 and 3 (e.g. Nano Lett. 2016, 16, 8, 5109–5113). Can all experimental observations be accounted for solely by this attractive interaction?

As discussed in our response to point 1 above, it is true that the bipolaron may break the symmetry and lead to some of the signal in the cross-polarized measurements; however, the coupling between the bipolaron and unbound two-polaron state is not trivial, the relative amplitude of the peaks precludes the possibility that the polaron cross peaks can be explained solely by the presence of the bipolaron; and the bipolaron cannot explain the presence of the diagonal peaks in the co-circular polarized measurements.

Actually, from a naïve picture, signal electric field in Fig. 3 is only contributed by S_{du} and T_{uu} that are spatially close, where short range interaction is likely more relevant (and certainly stronger). As an extreme example, if we separate the sample into two pieces and have E1, E3 only hitting one piece but E2 only hitting the other piece, the signal would be very small since at the end the transition matrix element between ground state in one piece and excited state in the other piece is small.

There appears to be a misunderstanding here. The separation of the S_{du} and T_{uu} is not relevant for determining the signal strength, what matters is the strength of the interaction, which can be attractive or repulsive.

In the case of a strong (infinite) repulsive interaction, where there is a reduced (zero) probability of exciting T_{uu} in the presence of S_{du} , and vice versa, then the response from the ESA pathway would be reduced (go to zero) while the SE response would remain. This dominant SE pathway leads to a signal being detected, which can be considered as arising due to the quasiparticles being coupled via a common ground state (as in the case for a 3-level system [e.g., 21,26,45,55]). With a strong (infinite) repulsion they must share a common ground state - even if the spatial separation is large: the fact that the presence of one is affecting the excitation of the other requires that they share a common ground state. Conversely, if the two quasiparticles are truly decoupled and not interacting, as would be the case in the example suggested by the reviewer, then they would not share a common ground state, and no signal would be expected. One could treat this as two uncoupled 2-level systems, which can be represented as a single four level system [e.g., 18,24]. In that case, since they are uncoupled and the presence of one does not affect the excitation of the other, the ESA and SE pathways in the effective 4-level system will once again cancel, to give zero signal. This comparison highlights the important distinction between a strong repulsive interaction between quasiparticles, which will lead to the generation of a signal, and two quasiparticles that are spatially separated and not interacting (or simply not interacting), which will not generate a signal. In our manuscript we consider repulsive interactions, where the strength of the measured signals are related to the strength of the interactions.

It is also worth clarifying the length scales associated with the interactions: When the distance between quasiparticles is on the order of the trion size, the competition for electrons leads to the exchange of electrons between excitons, resulting in an electron-mediated attraction and cooperative binding. However, when the quasiparticles are far apart, the competition for electrons results in an effective repulsion, since now only one of the excitons can interact with any given electron.

We have modified the text on p12 to clarify these length scales:

However, the effective repulsive interactions due to phase space filling are still compatible with the strong attractive interactions in the bipolaron state, since the former occurs for (unbound) polarons separated by large distances compared to the trion size, while the latter occurs at small polaron separation, on the order of the trion size, where the excitons can exchange an electron between each other.

3. Fig. 2 is very nice and worthy of more discussion. The manuscript states the conclusion with one sentence that “this indicates that interactions between polarons dressed by the same Fermi sea dominate”, which to my opinion is too simplified. For example, Fig. 2 offers strong evidence that interactions only happen between S_{du} and T_{uu} or T_{dd} and S_{ud} , but not between S_{ud} and T_{uu} or T_{uu} and S_{ud} . This conclusion is convincing. However, it can be explained by two scenarios: the fact that S_{ud} and T_{uu} do not interact could be either due to that the “d” fermi sea and the “u” fermi sea have different spins and do not interact, which is the picture used in the manuscript (Fig. 2b); or due to that two “u” exciton (i.e. two excitons from the same valley) strongly repel each other and therefore always stay far apart and do not interact. The absence of $S_{ud} \leftrightarrow S_{du}$ or $T_{ud} \leftrightarrow T_{du}$ in Fig. 2b actually suggests that indeed the interaction is still weak even if two excitons are from the different valleys, as long as the Fermi seas are different (as opposed to the exciton case in Nature Physics 12, 677–682 (2016)). On the other hand, this is still consistent with the fact that only two polarons with the same Fermi sea can form a bipolarons (Fig. 4b), and therefore can be accounted for by solely considering the short-range attractive interaction.

We thank the reviewer for the suggestion, and agree that Fig2 represents a key result and is worthy of more discussion. we have added the following text on p7:

“The presence of peaks on the diagonal for the co-circularly polarized excitation is indicative of strong interactions between identical polarons, while the absence of any cross-peaks implies there are no strong interactions between polarons consisting of identical excitons but dressed by electrons in opposite valleys. For the cross-circularly polarized pulses, the presence of cross-peaks and absence of diagonal peaks identifies strong interactions between polarons consisting of excitons in opposite valleys only when the interacting polarons are dressed by the same Fermi sea. Combined, these results show that regardless of the exciton valley there are strong interactions between polarons only when they are dressed by the same Fermi sea of electrons.”

The alternate scenario that the reviewer suggests (that the absence of interactions between S_{ud} and T_{uu} or T_{uu} and S_{ud} could be due to that two “u” excitons (i.e. two excitons from the same valley) strongly repel each other and therefore always stay far apart and do not interact) can be ruled out as follows:

- Firstly, there is a contradiction in the statement “strongly repel each other and therefore always stay far apart and do not interact”: such a strong repulsion is by definition a strong interaction and would actually lead to the presence of a signal as described above in response to point 1. (in the language of pump-probe spectroscopy, this relates to a ground state bleach, where the presence of T_{uu} reduces the probability of S_{ud} being excited) There is no such signal to indicate this type of interaction.
- Secondly, the fact that we observe a signal on the diagonal in the co-circular polarization measurement shows that there are interactions between polarons that have excitons in the same valley, provided the electrons that dress them are also in the same valley. (see discussion above in response to point 1 for an explanation of why these peaks require interactions)
- the final point that “The absence of $S_{ud} \leftrightarrow S_{du}$ or $T_{ud} \leftrightarrow T_{du}$ in Fig. 2b actually suggests that indeed the interaction is still weak even if two excitons are from the different valleys, as long as the Fermi seas are different” simply confirms our assertion that the polarons interact primarily through the Fermi sea electrons and hence only interact when they both involve the same Fermi sea in the same valley.
- In the Nat Phys paper referenced they observe interactions between excitons in opposite valleys, just as we do (see Fig S14(b)). They do not consider trion or polaron interactions, and there is no contradiction with our results.

A minor terminology issue: the authors use the first u/d to represent “spin of the exciton”, which is not accurate. All excitons discussed in the manuscript are bright and have total spin 0; the first u/d is representing spin of the electron within the exciton, not the exciton as a whole.

We thank the reviewer for picking this up, the total spin of the exciton is indeed 0. We have revised this to refer to the pseudospin of the exciton, which is directly related to the angular momentum of the light generating the excitons. We have added the following text and reference to introduce the pseudospin label, and modified all other references accordingly:

“... where \Uparrow (\Downarrow) represents the pseudospin [Xu, Nat Phys. 10,343,2014] of an exciton in the K (K^{\prime}) valley ...”

Reviewer #1 (Remarks to the Author):

Bose-polarons and Fermi-polarons are subject to intense experimental and theoretical research and the understanding of polaron-polaron interaction and the bipolaron structure is certainly a very important research direction. The authors use TMDCs which are particularly suited to the study of polaron-polaron interactions because one can create excitons in a controlled way and one can select the desired type of polarons by doping.

The paper is well written, clear, and thorough. The theoretical part uses a simple model and the good agreement with the experiments is somewhat surprising. But given the difficulty of the theoretical description of such a complex system, there is no suitable alternative approaches.

...

In overall, I have found this paper interesting and important contribution and recommend it for publication.

Reviewer #2 (Remarks to the Author):

In this manuscript, the authors report measurements of exciton-polaron interactions in monolayer WS₂ by multi-dimensional coherent spectroscopy. The results could be potentially useful to understand the complex interactions in TMDs. Before recommending this manuscript, I hope the authors could address the comments below.

Reviewer #3 (Remarks to the Author):

The manuscript by Muir et al. reports multi-dimensional coherent spectroscopy (MDCS) study of monolayer WS₂. Distinct signals between co- and cross polarized configurations are observed, indicating that interactions only happen between specific combinations of attractive polarons. The MDCS spectra also show additional features at lower energy, which are assigned to bipolarons. Overall, I find the results interesting. On the other hand, I am not fully convinced by the microscopic pictures in the manuscript.

Reviewers' Comments:

Reviewer #1:

Remarks to the Author:

The authors answered my comments and the paper is ready for publication.

Reviewer #2:

Remarks to the Author:

I'm appreciated of the authors' effort. The revised manuscript has addressed all my concerns. And I agreed with the change of the title to remove the ambiguity. I would thus recommend the publication of the manuscript in Nature Communications.

Reviewer #3:

Remarks to the Author:

The revised manuscript has largely addressed my comments. I can now recommend publication.